# Tooth discoloration caused by nanographene oxide as an irrigant and intracanal medicament in the endodontic treatment of extracted single-rooted teeth: An ex-vivo study

Abbas Abbaszadegan[1], Zeinab Rafiee[1]◉*, Bahar Asheghi[1]◉*, Ahmad Gholami[2]

**1** Department of Endodontics, School of Dentistry, Shiraz University of Medical Sciences, Shiraz, Iran,
**2** Biotechnology Research Center, School of Pharmacy, Shiraz University of Medical Sciences, Shiraz, Iran

◉ The authors contributed equally to this work.
* nobakhtf78@gmail.com (ZR); Bahar64a@gmail.com (BA)

## Abstract

Tooth discoloration caused by intracanal medicaments and irrigants presents a significant aesthetic challenge in dentistry. This study aimed to investigate the discoloration effects on tooth of nanographene oxide and positively charged silver nanoparticles and compares them with other commonly used materials in endodontic treatment. A total of 108 single-rooted, single-canal anterior mandibular and maxillary premolar teeth, extracted for orthodontic or periodontal reasons, were selected and prepared. The specimens were randomly divided into seven experimental groups and two control groups, each containing 12 samples. The experimental groups included three irrigants: nanographene oxide, sodium hypochlorite, and positively charged silver nanoparticles. The four medicament groups were nanographene oxide-carboxymethyl cellulose, positively charged silver nanoparticles-carboxymethyl cellulose, calcium hydroxide, and carboxymethyl cellulose. The control groups consisted of normal saline and blood. Discoloration was assessed at five time points: before material placement (T0), immediately after placement (T1), one week later (T2), one month later (T3), and three months later (T4). Spectrophotometric analysis was used to measure discoloration, and the ΔE values were statistically analyzed using repeated measures ANOVA and Tukey's post-hoc tests. After three months, no statistically significant difference in discoloration was observed among the irrigants (P > 0.05). However, the highest degree of discoloration was found in the silver nanoparticles group. Significant differences in ΔE values were noted between the normal saline group and both the silver nanoparticles (P = 0.001) and blood (P = 0.007) groups. Among the intracanal medicaments, a significant difference in ΔE values was observed between the carboxymethyl cellulose and calcium hydroxide groups (P = 0.005) at the final three-month examination. No significant differences were found among the other groups (P > 0.05). Nanographene oxide, used as both

**Data availability statement:** All relevant data are within the manuscript and its Supporting information files.

**Funding:** The Vice-Chancellery of Shiraz University of Medical Sciences for supporting this study (Grant #24674). The funders had no role in study design, data collection, analysis, decision to publish, or manuscript preparation.

**Competing interests:** No authors have competing interests.

an irrigant and medicament, does not cause more discoloration than other commonly used materials in endodontic treatment. Therefore, it can be considered a viable alternative to traditional endodontic materials.

## Introduction

Nanoparticles are materials defined by dimensions smaller than 100 nm, with their size and shape playing a critical role in influencing the physicochemical properties of substances [1]. These properties, in turn, affect how substances are absorbed and utilized. The use of nanoparticles enhances these properties, particularly their antimicrobial efficacy, which is further improved when functionalized nanoparticles are employed [2–4]. In endodontics, nanoparticles have found applications in areas such as tissue regeneration, drug delivery systems, and antimicrobial treatments, with the primary goal of improving oral health by targeting biofilms and bacteria [1,5]. The term 'nano-dentistry' was coined by Dr. Freitas Jr. in the year 2000. He developed nanomaterials and nanorobots, helped in regeneration of dentition, and developed dentifrobots – robots in dentrifices [6]. Nanodentistry is primarily focused on developing new materials or enhancing existing ones by incorporating nanomaterials that exhibit improved physical properties and functions [7]. These nanomaterials can mimic natural biological processes or act as nanorobots to facilitate more effective and precise dental treatments [8]. A wide range of nanostructures, such as nanoparticles (NPs), nanospheres, nanotubes, nanofibers, and nanorods, have been explored for dental applications, with silver nanoparticles (AgNPs) gaining particular attention due to their potent antimicrobial properties and biocompatibility at low concentrations [9]. AgNPs exert their antimicrobial effect by binding to microbial cell walls, disrupting cell membrane permeability, and ultimately leading to cell death by interfering with DNA and enzymatic functions [10,11]. Their rapid interaction with target cells, driven by a high surface area-to-volume ratio, also suggests their potential in reducing the risk of bacterial resistance [10,12].

While the use of nanoparticles, such as AgNPs, shows promise in disinfecting root canals, their potential adverse effects, particularly tooth discoloration, raise concerns about their suitability for clinical use, especially in long-term applications as irrigants and intracanal medicaments [13–15]. In recent years, carbon-based nanomaterials, specifically 2D materials like nanographene oxide (nGO), have garnered increasing interest due to their unique chemical, physical, and electronic properties [16]. In addition to nGO, other carbon-based nanomaterials include graphene quantum dots, carbon nanotubes, fullerenes, carbon dots, graphitic carbon nitride, mesoporous carbon, and nanodiamonds [17].

nGO, a derivative of graphene, has shown potential in a variety of dental applications, including drug delivery, antibacterial treatments, biocompatible scaffolds, and the enhancement of dental biomaterials [18]. Its distinctive properties—such as a high specific surface area (>1,000 m²/g), strong mechanical strength (Young's modulus of ~1,100 GPa), biocompatibility, low cost, and ease of functionalization—make

it an attractive candidate for dental research [19]. Additionally, nGO has a high affinity for metal ions, further expanding its potential applications in dentistry [20].

Despite these promising attributes, concerns regarding the potential for nGO to cause tooth discoloration must be addressed, as aesthetic considerations are critical in dental treatments [13–15]. Currently, no research has investigated the staining effects of nGO in endodontic procedures. Furthermore, there is a notable gap in studies examining the staining potential of nGO when utilized independently or in conjunction with other agents, both as an irrigant and as an intracanal medicament.

Given the importance of aesthetic outcomes in dentistry, it is essential to assess not only the biological and functional properties of nanomaterials but also their potential for causing undesirable side effects, such as discoloration [21].

It is noteworthy that the primary goal of endodontic treatment is to eliminate microbial contamination within the root canal system. However, complete removal of bacterial biofilm through mechanical instrumentation alone is often unachievable, particularly in the more complex regions of the canal system. As a result, intracanal medicaments and irrigants are utilized to disinfect these hard-to-reach areas [22,23]. Commonly used intracanal medicaments include calcium hydroxide $(Ca(OH)_2)$ and triple antibiotic paste, while sodium hypochlorite (NaOCl) and chlorhexidine (CHX) are among the most widely employed irrigants [7]. Despite their effectiveness, these traditional agents have several limitations, including cytotoxicity, changes in dentin structure, questionable long-term disinfection efficacy, and the potential to cause tooth discoloration [8,9]. These drawbacks have driven ongoing research into new materials and approaches aimed at overcoming these limitations and developing more ideal medicaments and irrigants.

This study aims to investigate the potential tooth discoloration caused by nGO at different time intervals and compare it to the discoloration effects of commonly used intracanal medicaments in endodontic treatment. This research seeks to provide a comprehensive evaluation of nGO's viability in an ex vivo model before clinical applications, balancing its functional benefits with aesthetic considerations.

## Materials and methods

This study was approved by the Ethics Committee of Shiraz University of Medical Sciences (Approval No: IR.SUMS.DENTAL.REC.1400.120). The study utilized 108 single-rooted, single-canal mandibular and maxillary premolar teeth extracted for orthodontic or periodontal reasons between June 3, 2022, and December 25, 2022. The sample size was determined based on a previous calculation using G*Power 3.1 software. Teeth were extracted at the surgery ward of the School of Dentistry, Shiraz University of Medical Sciences, and informed consent was obtained from all patients.

### Preparation of teeth

All teeth included in the study had intact crowns (without caries or fractures), mature apices, and no cervical abrasion were included in the study. Teeth were examined for cracks and resorption under a microscope (Zeiss, Oberkochen, Germany). Surface debris and calculus were removed using a scaler (Supprasson, Satelec, France), pumice paste, and a rubber cup. Periapical radiographs were taken from two angles to confirm the presence of a single canal. To standardize root length, 2–4 mm of the apex was removed. The samples were stored in sealed test tubes at 100% humidity for two weeks by one operator and then the root canals were instrumented from the apex to the most coronal part of the pulp chamber using ProTaper rotary files (S1, S2, F1, F2, F3; Dentsply Maillefer, Ballaigues, Switzerland) following the manufacturer's instructions. Gates Glidden drills (sizes 1–6; Mani, Tochigi, Japan) were used to complete canal preparation. During instrumentation, 5 mL of 2.5% NaOCl was used as an irrigant. To remove residual pulp tissue, 5.25% NaOCl was activated using a #3 ultrasonic tip (ProUltra, Dentsply, Johnson City, TN, USA) connected to an ultrasonicator (Supprasson, Satelec, France).

To remove the smear layer and open the dentinal tubules, canals were rinsed with 17% EDTA (Aria Dent, Tehran, Iran) for two minutes. The canals were dried with paper points, and 5 mL of the test solution was introduced into each canal

for 30 minutes, with replacement as needed. Finally, canals were irrigated with 5 mL of normal saline using a 27-gauge syringe, and the apical portion of each canal was sealed with restorative glass ionomer (Fuji Corporation, Tokyo, Japan). The samples were stored in boxes with 100% humidity at 37°C in an incubator. To prevent displacement, the teeth were mounted in silicone molds (Figs 1–3).

## Synthesis of nGO

nGO was synthesized using a modified version of Hummers' method as described by Eskandari et al. [24,25]. In this process, 1 g of graphite powder was dissolved in 23 mL of 98% $H_2SO_4$ (v/v) and stirred in a three-neck flask for three days. The solution was placed in an ice-water bath, and 6 g of $KMnO_4$ was added gradually, turning the solution from black to dark green. The mixture was transferred to an oil bath at 40°C, stirred for 30 minutes, and heated to 70°C for 45 minutes. Afterward, 6 mL of distilled water was added, and the mixture was heated to 105°C for 10 minutes. An additional 40 mL of distilled water was then added and heated to 100°C for 15 minutes. The reaction was terminated by adding 150 mL of distilled water and 15 mL of 35% $H_2O_2$ solution, turning the solution yellow-brown. The product was centrifuged at 10,000 g for 5 minutes, and the precipitate was washed twice with 5% HCl (v/v) and five times with distilled water. The final nGO

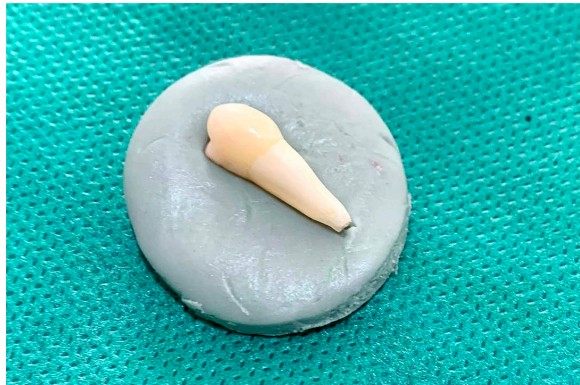

**Fig 1. The teeth were mounted in silicone molds.**

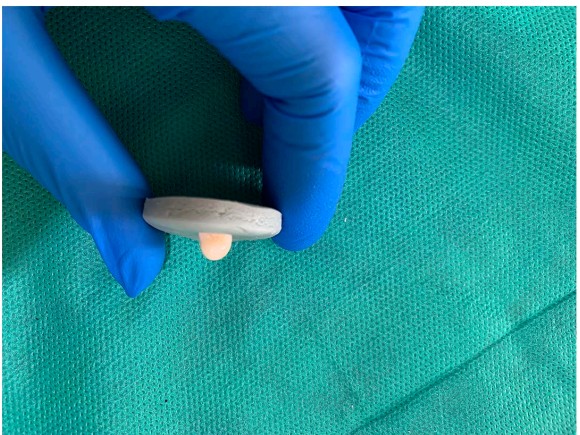

**Fig 2. The teeth were mounted in silicone molds.**

   

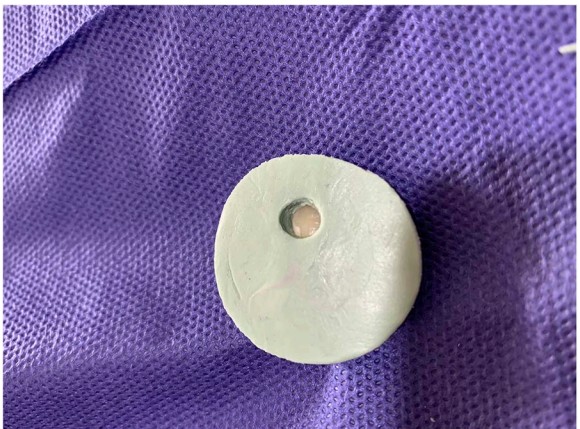

**Fig 3. The teeth were mounted in silicone molds.**

solution was prepared at a concentration of 1 mg/mL and used as an irrigant. The nGO was mixed with carboxymethyl cellulose (CMC) to prepare the medicament.

### Synthesis of imidazolium-coated silver nanoparticles (AgNp⁺)

AgNp⁺ were synthesized according to the method described by Gholami et al. [10]. Specifically, 0.1 mL of aqueous AgNO3 (Sigma Aldrich, USA) was stirred with 6.2 mM 1-dodecyl-3-methylimidazolium. A 0.4 M aqueous solution of NaBH4 was added dropwise until a golden color appeared. The solution was centrifuged for 20 minutes to remove residual ionic liquids. The final product, a 1 mg/mL solution of AgNp⁺, was used as an irrigant. The silver nanoparticle precipitate was also mixed with CMC for medicament preparation.

### Preparation of irrigants

Teeth were randomly assigned to three groups of 12 samples each, based on the irrigant used: AgNp⁺, nGO, and NaOCl. Blood was used as a positive control, and normal saline as a negative control.

### Preparation of intracanal medicaments

Forty-eight teeth, similar in characteristics to those in the irrigant group, were selected for intracanal medicament testing. Standard access cavities were prepared using a fissure bur (d&z, Wiesbaden, Germany) and a round bur (Dentsply Maillefer, Ballaigues, Switzerland). The root canals were instrumented from the coronal part of the root to the apex using ProTaper rotary files (S1, S2, F1, F2, F3; Dentsply Maillefer, Ballaigues, Switzerland) following the manufacturer's instructions. Gates Glidden drills (sizes 1–6; Mani, Tochigi, Japan) were used to complete canal preparation. The teeth were stored in physiological saline until use. After canal preparation, a final irrigation with 5 mL of saline was performed. Once the canals were dried with paper points, a standardized amount of medicament was placed below the cementoenamel junction (CEJ) using a Lentulo spiral. The teeth were then sealed with restorative glass ionomer. Samples were divided into four groups (12 per group):

- Group 1: CaOH$_2$ (Golchai Dent, Iran)

- Group 2: AgNp⁺-CMC (Merck, Germany)

- Group 3: GONps-CMC

- Group 4: CMC (Afa Chemi, Tehran, Iran)

  For medicaments containing nGO and AgNp$^+$, the materials were mixed with CMC.

## Evaluation of Tooth Discoloration

Tooth discoloration was measured using the VITA Easyshade Compact spectrophotometer (Vita Zahnfabrik, Bad Säckingen, Germany) to ensure accuracy and reproducibility. Measurements were taken at five stages:

- T0: Before material application

- T1: Immediately after application

- T2: One week post-application

- T3: One month post-application

- T4: Three months post-application

   Measurements were conducted by a technician in a controlled lighting environment, 3 mm above the CEJ on the buccal surface. Each measurement was repeated three times, and the mean value was recorded. All steps were performed by one operator. Color changes were assessed using the CIE Lab* color system [26], and the difference in discoloration (ΔE*) between two time points was calculated using the formula:

$$\Delta E* = [(L1 - L0*)^2 + (a1 - a0*)^2 + (b1 - b0*)^2]^{½}$$

## Statistical analysis

The Kolmogorov-Smirnov test was used to evaluate the normality of color difference data. Tukey's post-hoc test and ANOVA were employed to compare the discoloration caused by different materials, with statistical significance set at $p < 0.05$. All analyses were performed using IBM SPSS Statistics version 26.

## Results and discussion

All the irrigants used in this study caused immediate tooth discoloration upon placement inside the canal, and this discoloration persisted throughout the three-month observation period (Fig 4).

   Based on the ΔE values, all materials exhibited some degree of discoloration over the three months. Among the tested materials, normal saline resulted in the least discoloration, while AgNp$^+$ caused the most. A significant difference in ΔE values was observed between the normal saline group and both the AgNp$^+$ ($p = 0.00$) and blood ($p = 0.00$) groups. However, no statistically significant differences were found between the other groups ($p > 0.05$). The levels of discoloration at different time points are presented in Table 1.

   Figs 5 and 6 show the mean and standard deviation of tooth discoloration over the three-month follow-up period for the two nanoparticles used as irrigants.

   Similarly, the intracanal medicaments also caused immediate tooth discoloration, which persisted over the three months of follow-up (Fig 7). According to the ΔE values, all medicaments caused discoloration throughout the study period. CMC produced the least discoloration, while Ca(OH)$_2$ resulted in the greatest degree of discoloration. A significant difference in ΔE values was observed between the CMC and Ca(OH)$_2$ groups ($p = 0.00$), but no significant differences were found among the other groups ($p > 0.05$). The levels of discoloration at different time points are detailed in Table 2.

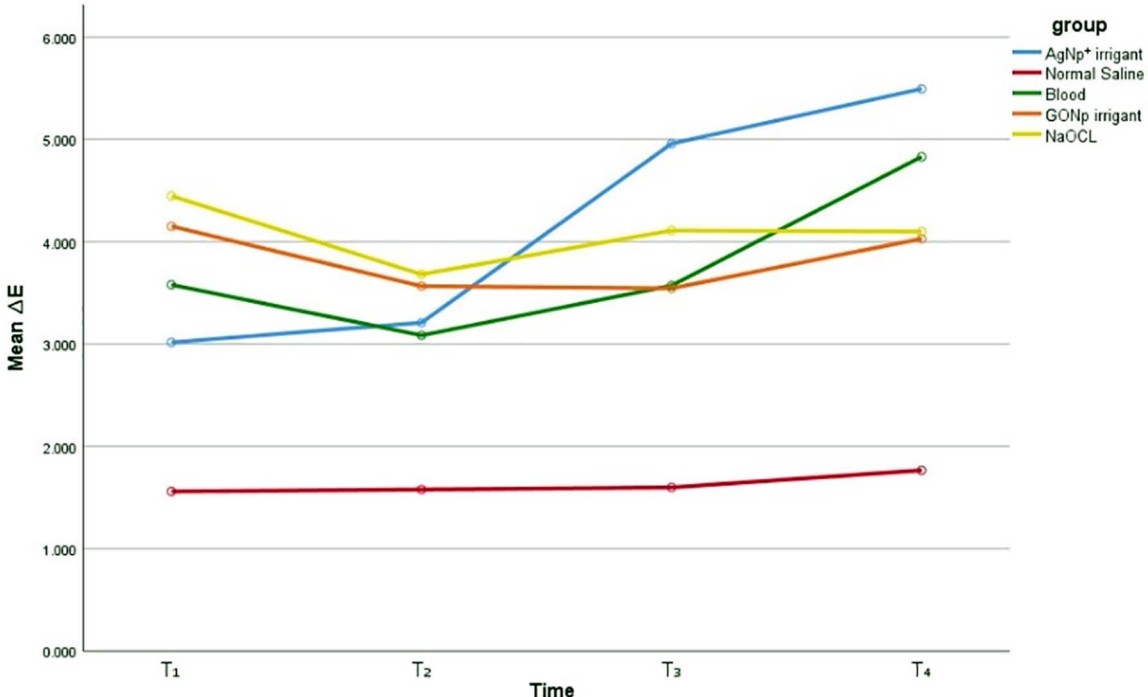

**Fig 4. The mean tooth discolorations after using canal irrigants at different periods.**

**Table 1. The mean and standard deviation of the discolorations of canal irrigants.**

| Group (subgroup) | *$T_1$ | **$T_2$ | ***$T_3$ | ****$T_4$ |
|---|---|---|---|---|
| Normal saline | 1.56±0.54^A | 1.58±0.061^A | 1.60±0.41 ^AC | 1.77±0.55^A |
| AgNp⁺ irrigant | 3.02±2.41 ^AC | 3.21±2.28 ^AC | 4.96±2.43^BC | 5.49±3.13^BC |
| GONp irrigant | 4.15±2.4^BC | 3.57±1.60 ^AC | 3.54±1.94 ^AC | 4.03±1.99 ^AC |
| NaOCL | 4.45±3.19^BC | 3.68±2.25^BC | 4.11±2.16^BC | 4.10±2.51 ^AC |
| Blood | 3.58±1.79 ^AC | 3.08±1.66 ^AC | 3.57±1.23 ^AC | 4.83±1.52^BC |

*T1 (immediately after being placed inside the canal.), **T2 (one week later), ***T3 (one month later), and ****T4 (three months later). Having the same uppercase English letters in each vertical column indicates that there are no statistically significant differences between the groups ($p<0.05$).

## Periods

Figs 8 and 9 present the mean and standard deviation of tooth discoloration for the two nanoparticles used as medicaments over the three-month follow-up period.

Nano-irrigants such as nGO and AgNp⁺ have recently emerged as promising materials for endodontic treatments. However, their potential to cause dentin discoloration, due to the presence of metallic ions in their structures, raises concerns about aesthetic outcomes. Understanding this issue is crucial for clinicians to make informed decisions when selecting nanomaterials, ensuring both effective treatment and satisfactory aesthetics [27–29].

Endodontic materials must maintain chromatic stability to prevent long-term discoloration [30]. Despite the growing interest in nano-irrigants, research on their long-term color stability is limited. This study is the first to investigate discoloration caused by nGO as both an irrigant and an intracanal medicament, comparing it to AgNp⁺ and other commonly used

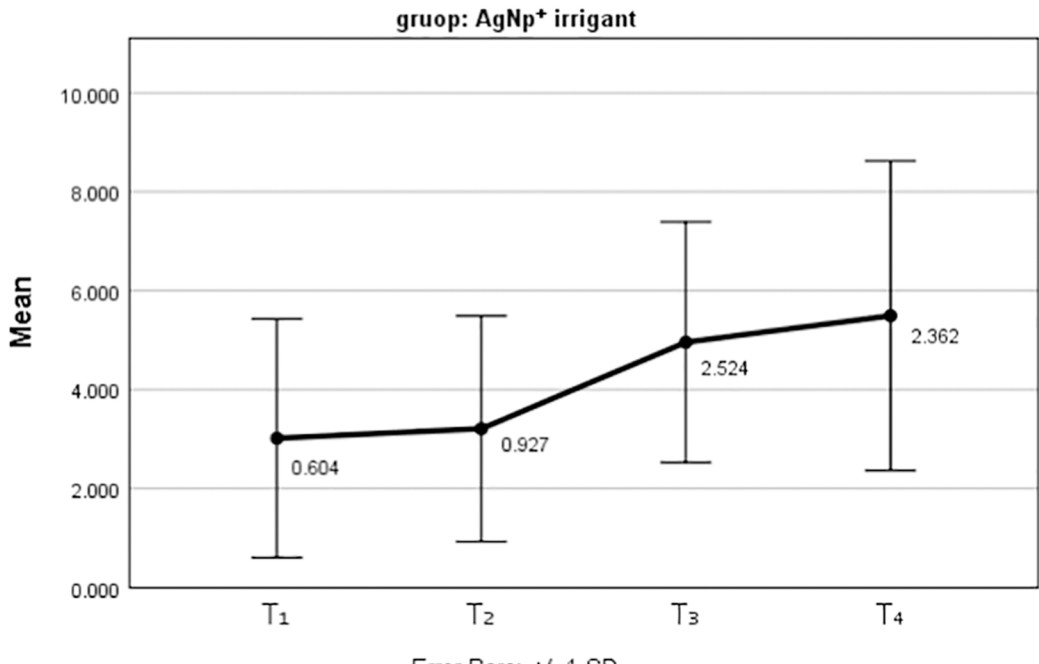

**Fig 5. The mean tooth discolorations after using AgNp⁺ irrigants at different periods.**

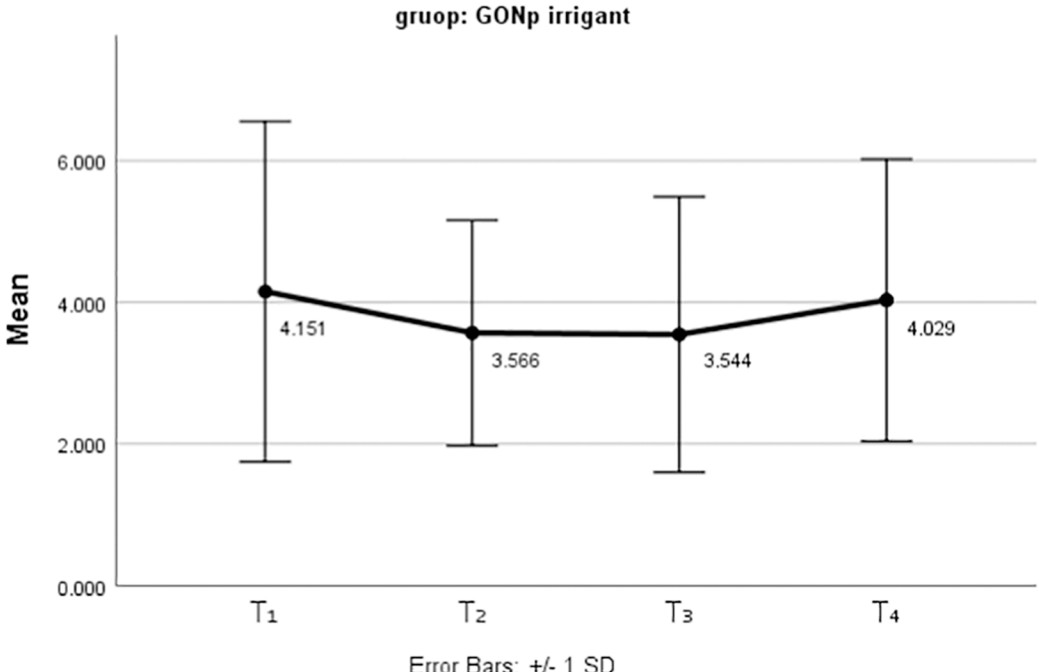

**Fig 6. The mean tooth discolorations after using GONp irrigants at different periods.**

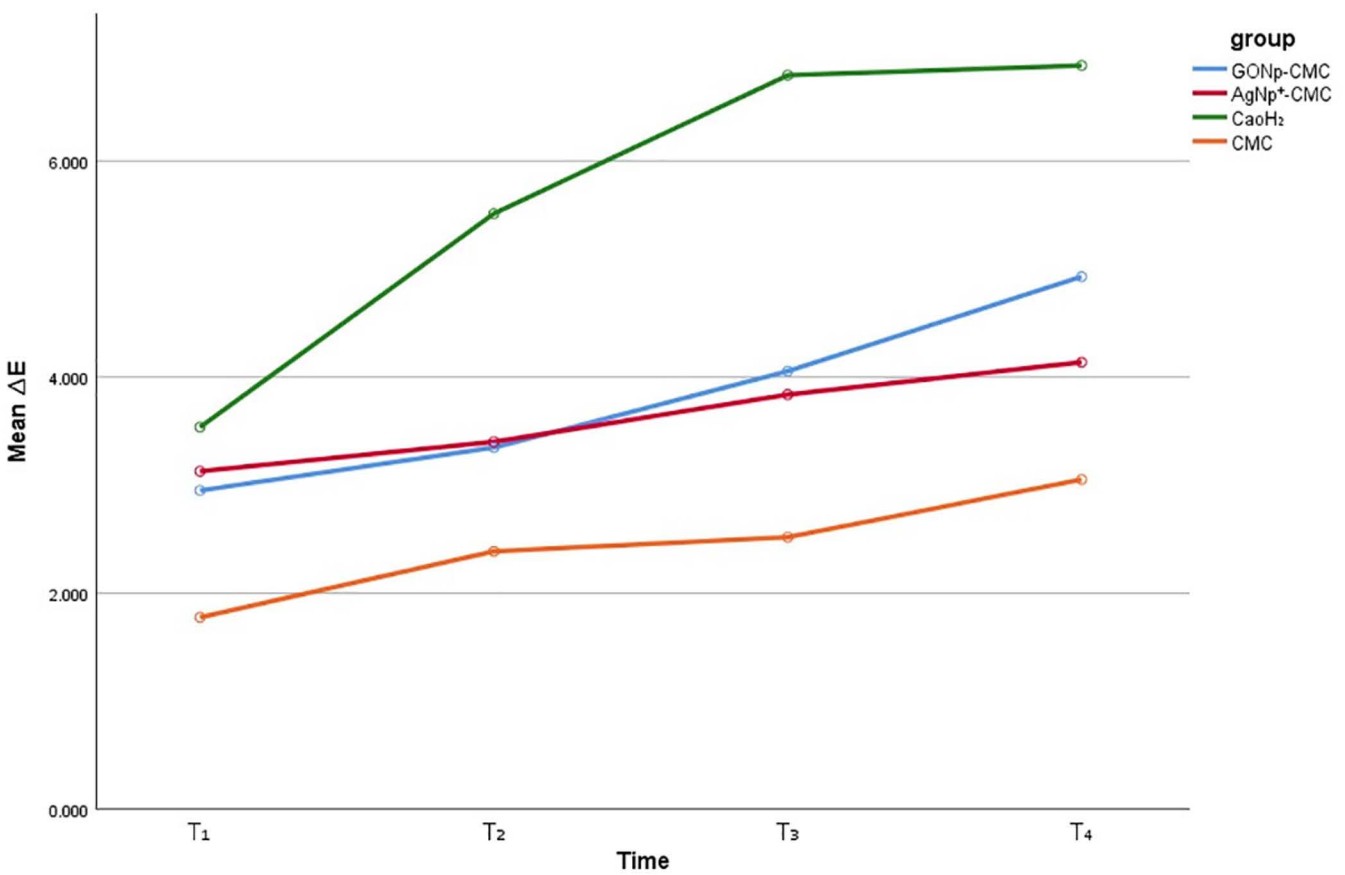

**Fig 7. The mean tooth discolorations after the use of intracanal medicaments at different.**

**Table 2. The mean and standard deviation of the discolorations of the intracanal medicament.**

| Group (subgroup) | *T₁ | **T₂ | ***T₃ | ****T₄ |
|---|---|---|---|---|
| GONp-CMC | 2.95±1.37ᴬ | 3.35 ±1.89 ᴬᶜ | 4.05±2.24 ᴬᶜ | 4.93±2.57 ᴬᶜ |
| AgNP⁺-CMC | 3.13±1.97ᴬ | 3.40±.1.95 ᴬᶜ | 3.84±2.43ᴬ | 4.14±3.42 ᴬᶜ |
| CaOH₂ | 3.54±2.31ᴬ | 5.51±2.22ᴮ | 6.80±3.26ᴮᶜ | 6.89±2.86ᴮᶜ |
| CMC | 1.78±1.03ᴬ | 2.39±0.75ᴬ | 2.52±1.25ᴬ | 3.05±1.12ᴬ |

*T1 (immediately after being placed inside the canal.), **T2 (one week later), ***T3 (one month later), and ****T4 (three months later). Having the same uppercase English letters in each vertical column indicates no statistically significant differences between the groups (p<0.05).

endodontic materials. In line with previous research, this study utilized a spectrophotometer to measure discoloration. Spectrophotometers are highly sensitive and can detect even minor color changes imperceptible to the naked eye, making them superior for assessing tooth discoloration [31,32].

To simulate clinical conditions more accurately, human teeth were used in this study. The irrigant groups underwent cleaning and preparation through the apex, creating a sealed environment, while the medicament groups had access created through the crown to allow placement of materials below the CEJ, following established methodologies [31,33–35]. Blood served as a positive control due to its well-known capacity to induce tooth discoloration [36–38].

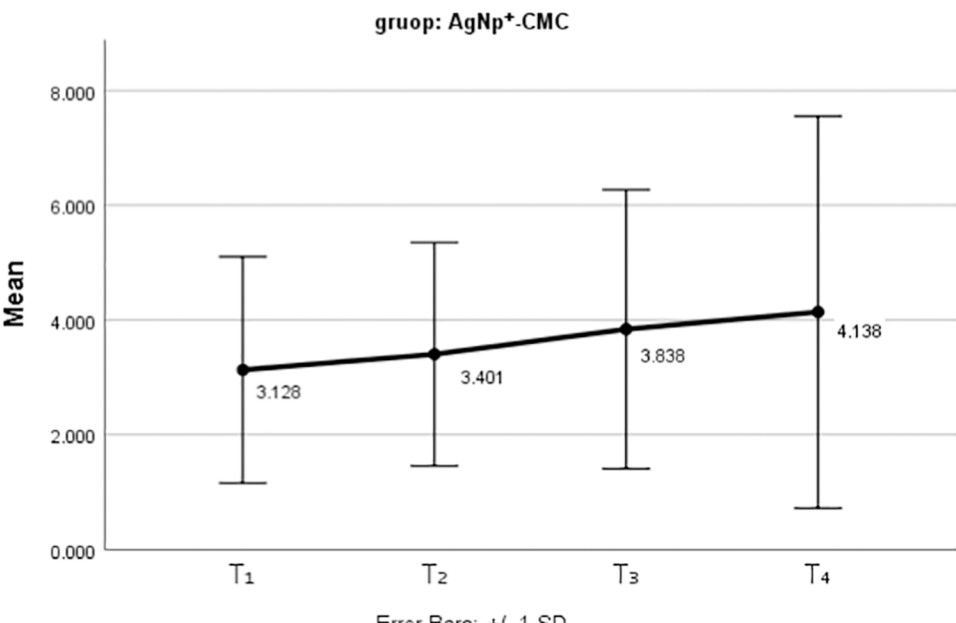

**Fig 8. The mean tooth discolorations after the use of AgNP+- CMC as an intracanal medicament at different periods.**

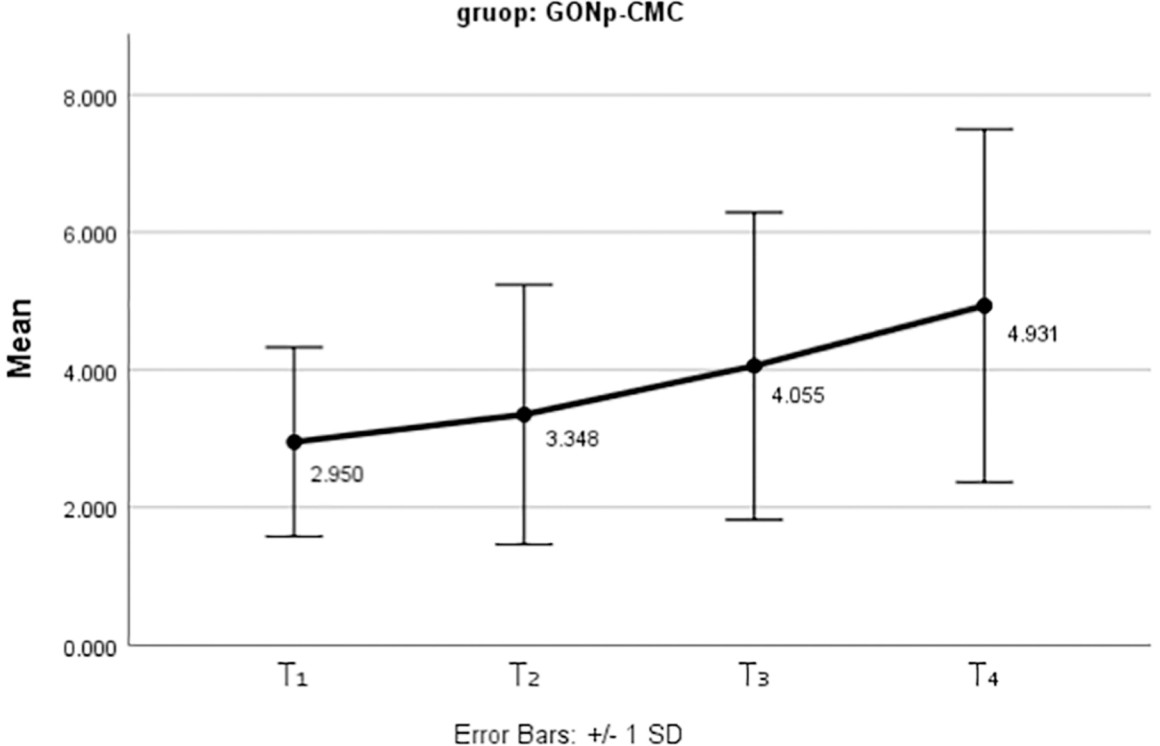

**Fig 9. The mean tooth discolorations after the use of GONp-CMC as an intracanal medicament at different periods.**

Removing the smear layer was a crucial step, as it enhances the penetration of medicaments into dentinal tubules. The smear layer affects dentin permeability and can prolong discoloration [33]. CMC was used as a carrier for nGO and AgNp+, as it improves the delivery and interaction of these medicaments with dentin surfaces [39].

Our results showed significant discoloration with AgNp+ when used as an irrigant after one month, with these changes persisting for up to three months, consistent with previous studies [34,40]. The silver ions likely contribute to discoloration through reduction or oxidation reactions when exposed to dentin collagen and NaOCl. Additionally, AgNPs may cause discoloration over time by replacing calcium ions in dentin with metal ions released from the nanoparticles [41].

nGO, a derivative of graphene, was chosen for its hydrophilic, biocompatible properties, along with its strong antimicrobial and regenerative potential [41]. Unlike silver nanoparticles, nGO did not cause significant discoloration compared to normal saline. While some discoloration was observed, it remained relatively minor across all time intervals. The minimal discoloration may be attributed to the tubule-sealing properties of nGO, which may prevent the penetration of discoloring agents into the dentin [42].

When comparing intracanal medicaments, most materials caused progressively increasing discoloration over three months, showing slightly different behavior compared to their role as irrigants. Interestingly, while nGO and AgNP+ did not cause significant color changes compared to $Ca(OH)_2$, AgNP+ consistently exhibited lower ΔE values than nGO after three months. This suggests that AgNP+ may have a lower clinical impact in terms of perceived chromatic alterations, making it a more favorable choice as an intracanal medicament.

Both nGO and AgNP+, when pre-mixed with CMC, exhibited less discoloration over time than $Ca(OH)_2$. The use of CMC as a carrier may reduce interactions between these materials and the surrounding dentin, contributing to their lower discoloration potential. However, differences in adhesion between CMC and dentin may still result in some discoloration over time.

$Ca(OH)_2$ showed the most significant discoloration, with ΔE values indicating a noticeable color change perceptible to the human eye. These results are consistent with other studies, such as Afkhami et al., who reported similar findings with $Ca(OH)_2$ [31].

Finally, our study highlights the potential of nGO as a novel material in endodontics, combining strong antimicrobial properties with minimal aesthetic drawbacks. It has been shown to inhibit the growth of several oral pathogens and is non-toxic to human dental pulp stem cells and epithelial cells [42,43]. Additionally, we have demonstrated that its discoloration potential is comparable to other commonly used materials in endodontics. However, future studies should investigate the mechanisms of discoloration in greater detail and explore the effects of different concentrations and application methods on long-term color stability.

## Conclusions

Despite the limitations of this study, including its ex vivo design and short follow-up period, nGO demonstrated a similar discoloration potential to the other materials tested, with no significant influence of application method or timing on the degree of discoloration. Further research is needed to explore the mechanisms behind nGO's interaction with dentin and its long-term effects. Additionally, studies on bleaching methods may help improve the clinical applicability of nGO in aesthetic dental treatments.

## Supporting information

**S1 Table. The mean and standard deviation of the discolorations of canal irrigants.** *T1 (immediately after being placed inside the canal.), **T2 (one week later), ***T3 (one month later), and ****T4 (three months later). Having the same uppercase English letters in each vertical column indicates that there are no statistically significant differences between the groups (p < 0.05).
(DOCX)

**S2 Table. The mean and standard deviation of the discolorations of the intracanal medicament.** *T1 (immediately after being placed inside the canal.), **T2 (one week later), ***T3 (one month later), and ****T4 (three months later). Having the same uppercase English letters in each vertical column indicates no statistically significant differences between the groups (p < 0.05).
(DOCX)

## Acknowledgments

The authors would like to express their appreciation to Dr. Alireza Nobakht for the English language editing of this paper and Dr. Ahmad Baseri for his assistance with statistical analyses.

## Author contributions

**Conceptualization:** Bahar Asheghi, Ahmad Gholami.

**Data curation:** Zeinab Rafiee.

**Formal analysis:** Zeinab Rafiee.

**Funding acquisition:** Abbas Abbaszadegan.

**Investigation:** Zeinab Rafiee.

**Methodology:** Zeinab Rafiee, Ahmad Gholami.

**Project administration:** Ahmad Gholami.

**Resources:** Bahar Asheghi.

**Supervision:** Abbas Abbaszadegan, Bahar Asheghi, Ahmad Gholami.

**Validation:** Abbas Abbaszadegan, Ahmad Gholami.

**Visualization:** Abbas Abbaszadegan.

**Writing – original draft:** Abbas Abbaszadegan, Zeinab Rafiee, Bahar Asheghi.

**Writing – review & editing:** Abbas Abbaszadegan.

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
