## [Decision Letter · Decision Letter 0]

Dear Dr. Rafiee,

Thank you for submitting your manuscript to PLOS ONE. After careful consideration, we feel that it has merit but does not fully meet PLOS ONE’s publication criteria as it currently stands. Therefore, we invite you to submit a revised version of the manuscript that addresses the points raised during the review process.

Dear Authors,

Kindly read all the comments given by the reviewers carefully and address them; make the changes in the revised manuscript accordingly.

Best regards and keep well

We look forward to receiving your revised manuscript.

Kind regards,

Mohmed Isaqali Karobari, BDS, MScD.Endo, Ph.D. Endo, FDS RCS (Eng), FPFA, MFDS RCGS (Glasg)

Academic Editor

PLOS ONE

Journal Requirements:

"The Vice-Chancellery of Shiraz University of Medical Sciences for supporting this study (Grant #24674)"

"This manuscript is based on the postgraduate thesis by Dr. Zeinab Rafiee. The authors would like to express their appreciation to the Vice-Chancellery of Shiraz University of Medical Sciences for supporting this study (Grant #24674). We also thank Dr. Alireza Nobakht for the English language editing of this paper and Dr. Ahmad Baseri for his assistance with statistical analyses."

"The Vice-Chancellery of Shiraz University of Medical Sciences for supporting this study (Grant #24674)"

"NO authors have competing interests"

6. Please provide a complete Data Availability Statement in the submission form, ensuring you include all necessary access information or a reason for why you are unable to make your data freely accessible. If your research concerns only data provided within your submission, please write "All data are in the manuscript and/or supporting information files" as your Data Availability Statement.

Additional Editor Comments:

Dear Authors,

Kindly read all the comments given by the reviewers carefully and address them; make the changes in the revised manuscript accordingly.

Best regards and keep well

Reviewers' comments:

Reviewer's Responses to Questions

**Comments to the Author**

1. Is the manuscript technically sound, and do the data support the conclusions?

Reviewer #1: Yes

Reviewer #2: Yes

2. Has the statistical analysis been performed appropriately and rigorously?

Reviewer #1: No

Reviewer #2: Yes

3. Have the authors made all data underlying the findings in their manuscript fully available?

Reviewer #1: Yes

Reviewer #2: Yes

4. Is the manuscript presented in an intelligible fashion and written in standard English?

Reviewer #1: Yes

Reviewer #2: Yes

Reviewer #1: 1. Sample size calculation needs to be added in Materials and methods with relevant references.

2. Results needs to be explained further in detail.

3. Tables needs to be more detailed by adding the headers for columns- more clarity.

4. Other comments are highlighted in the manuscript changes in the presentation and grammar errors

Reviewer #2: Dear Authors,

I have reviewed your original study and have a few comments for improvement. Please follow the points below:

Abstract:

The abstract is well written.

Keywords:

Revise the keywords and include at least six MeSH terms related to your work.

Introduction:

The introduction needs to be updated with the latest information and enriched with the best literature to support this study and enhance readability. I suggest including the following references in the first introductory paragraph:

a) Hayat, Fatima, et al. "Nanoparticles in endodontics." Biomaterials in Endodontics (2022): 195-209.

b) Khurshid, Z., Alqurashi, H., & Ashi, H. (2024). Advancing Environmental Sustainability in Dentistry and Oral Health. European Journal of General Dentistry.

Discuss the sustainability of your new work and how it opens avenues for future sustainable dental biomaterials.

In the paragraph "The most common intracanal medicaments are triple antibiotic paste and calcium hydroxide (Ca (OH)2), whereas the most common irrigants... questionable disinfection ability, and tooth discoloration have led studies toward removing these limitations in order to obtain ideal materials," please provide original studies to support this statement. This critique sets the stage for your original study.

To support the statement "Nanoparticles (with a diameter of ≤100 nm) have become popular antimicrobial... Their larger charge density, surface area, and area lead to more interaction with bacteria," you can cite the following papers:

Tekin, Bahar, et al. "Effect of micro-arc oxidation coatings with graphene oxide and graphite on osseointegration of titanium implants—an in vivo study." The Saudi Dental Journal 36.4 (2024): 591-595.

Najeeb, Shariq, et al. "Dental applications of nanodiamonds." Science of Advanced Materials 8.11 (2016): 2064-2070.

Khurshid, Z., Zafar, M., Qasim, S., Shahab, S., Naseem, M., & AbuReqaiba, A. (2015). Advances in nanotechnology for restorative dentistry. Materials, 8(2), 717-731.

Methods and Methodology:

Mention the city and country in the statement "These teeth were referred for extraction to the surgery ward of Shiraz Dental School."

Provide a sample figure in the "Preparation of the teeth" section.

The materials preparation section is not detailed enough. Instead of "It should be noted that for the preparation of medicaments containing nanographene oxide and silver-imidazolium nanoparticles, each of these materials was mixed," there should be a separate heading for material preparation with detailed steps.

How did you check the esthetic of the prepared tooth after intervention? Did you use any esthetic scale to validate this?

Results and Discussion:

The results and discussion sections need careful revision. The writing is weak, and the authors often link previous work by merely citing author names without explaining the previous work in their own words.

Conclusion:

The conclusion is not well written. Revise the conclusion to better summarize the study's findings and include future prospects.

Thank you for considering these suggestions. I believe addressing these points will significantly improve the quality and impact of your study.

Best regards

**Do you want your identity to be public for this peer review?** For information about this choice, including consent withdrawal, please see our Privacy Policy

Reviewer #1: No

Reviewer #2: No

---

## [Author Response · Author response to Decision Letter 1]

13 Oct 2024

PONE-D-24-30140

Tooth discoloration caused by nanographene oxide as an irrigant and intracanal medicament in the endodontic treatment of extracted single-rooted teeth: an ex-vivo study

Dear Editor,

We would like to express our sincere gratitude for your valuable comments, which have significantly helped improve the quality of our manuscript. In response, we have thoroughly revised the manuscript and addressed all the concerns raised by the reviewers in a point-by-point manner. The changes have been highlighted in the revised manuscript. Please find below our responses to the reviewers’ comments.

Kind regards,

Zeinab Rafiee

1. PLOS ONE Style Requirements

Comment: Please ensure that your manuscript meets PLOS ONE’s style requirements, including those for file naming. The PLOS ONE style templates can be found at:

Response:

We have adjusted the reference formatting and ensured that our manuscript now adheres to all PLOS ONE style requirements, including those for file naming.

2. Grant Information Discrepancy

Comment: The grant information provided in the ‘Funding Information’ and ‘Financial Disclosure’ sections do not match. Please ensure the correct grant numbers are provided in the ‘Funding Information’ section.

Response:

The grant number and ethics approval number are distinct as per our institutional policy. To avoid confusion, we have selected and included the appropriate code in the ‘Funding Information’ section.

3. Financial Disclosure and Role of Funders

Comment: Please state the role of the funders in your study. If they had no role, include the statement: “The funders had no role in study design, data collection and analysis, decision to publish, or preparation of the manuscript.”

Response:

We confirm that the funders had no role in study design, data collection and analysis, decision to publish, or preparation of the manuscript. This statement has been added to the cover letter, and we kindly ask you to update the online submission form accordingly.

4. Funding in Acknowledgments Section

Comment: Please remove any funding-related text from the Acknowledgments section. Funding information should only appear in the Funding Statement.

Response:

We have removed the grant number and funding-related text from the manuscript body and Acknowledgments section as per your policy. The Funding Statement has been updated accordingly.

5. Competing Interests

Comment: Please declare any Competing Interests. If there are none, state: “The authors have declared that no competing interests exist.”

Response:

We have declared that no competing interests exist, and this information has been provided in the cover letter. Please update the online submission form accordingly.

6. Data Availability Statement

Comment: Please provide a complete Data Availability Statement in the submission form.

Response:

We have provided a complete Data Availability Statement in the submission form, stating: “All data are included in the manuscript and/or supporting information files.”

Reviewer Comments:

Reviewer #1:

1. Sample Size Calculation

Comment: Please add a sample size calculation in the Materials and Methods section with relevant references.

Response:

The sample size used in this study was based on a previous calculation performed using G*Power 3.1 software, and this has been added to the Materials and Methods section.

2. Results Clarification

Comment: Please explain the Results section in more detail.

Response:

We have expanded the Results section to provide greater clarity.

3. Tables Enhancement

Comment: Tables need to be more detailed, including headers for columns to improve clarity.

Response:

The tables have been revised to include headers and provide a clearer presentation of the data. Statistical significance is indicated for each group and time point.

4. Grammar and Presentation

Comment: Other comments regarding grammar and presentation have been highlighted in the manuscript.

Response:

All grammatical and presentation-related comments have been addressed and revised in the manuscript.

Reviewer #2:

1. Keywords Revision

Comment: Revise the keywords to include at least six MeSH terms.

Response:

The keywords have been revised as suggested, and now include six MeSH terms related to the study.

2. Introduction Update

Comment: The introduction should be updated with the latest information, including references to recent literature.

Response:

The introduction has been updated for better readability and to incorporate the suggested references. We have also included information on the sustainability of this work, which opens new avenues for future research in sustainable dental biomaterials.

3. Supporting Literature for Statements

Comment: Provide original studies to support specific statements in the Introduction.

Response:

We have cited the recommended original studies to support the statements regarding commonly used irrigants and nanoparticles, enhancing the scientific foundation of the Introduction.

4. Material Preparation and Esthetic Validation

Comment: Provide more details on the materials preparation and esthetic validation methods.

Response:

Two new subheadings have been added to the Materials and Methods section to detail the preparation of the irrigants and medicaments. Esthetic validation was performed using the VITA Easyshade Compact spectrophotometer, and this information has been included in the manuscript.

5. Results and Discussion Revision

Comment: The Results and Discussion sections need to be revised for clarity, and the linkage to previous work should be explained more clearly.

Response:

Both the Results and Discussion sections have been extensively rewritten for clarity, and connections to previous work are now better explained.

6. Conclusion Revision

Comment: The conclusion needs to be revised to better summarize the study’s findings and future prospects.

Response:

The Conclusion section has been revised to more effectively summarize the findings and highlight potential future directions in dental biomaterials research.

Thank you for considering our revised manuscript. We look forward to your feedback.

Best regards,

Zeinab Rafiee

---

## [Decision Letter · Decision Letter 1]

Dear Dr. Rafiee,

Thank you for submitting your manuscript to PLOS ONE. After careful consideration, we feel that it has merit but does not fully meet PLOS ONE’s publication criteria as it currently stands. Therefore, we invite you to submit a revised version of the manuscript that addresses the points raised during the review process.

Dear Authors,

Kindly read all the comments given by the reviewers carefully and address them; make the changes in the revised manuscript accordingly.

Best regards and keep well

We look forward to receiving your revised manuscript.

Kind regards,

Mohmed Isaqali Karobari, BDS, MScD.Endo, Ph.D. Endo, FDS, FPFA, MFDS

Academic Editor

PLOS ONE

Journal Requirements:

Additional Editor Comments:

Dear Authors,

Kindly read all the comments given by the reviewers carefully and address them; make the changes in the revised manuscript accordingly.

Best regards and keep well

Reviewers' comments:

Reviewer's Responses to Questions

**Comments to the Author**

Reviewer #1: All comments have been addressed

2. Is the manuscript technically sound, and do the data support the conclusions?

Reviewer #1: Yes

3. Has the statistical analysis been performed appropriately and rigorously?

Reviewer #1: Yes

4. Have the authors made all data underlying the findings in their manuscript fully available?

Reviewer #1: Yes

5. Is the manuscript presented in an intelligible fashion and written in standard English?

Reviewer #1: Yes

Reviewer #1: All previous comments have been addressed by the author.

1. The reference for Evaluation of Tooth discoloration needs to be added in the methodology.

**Do you want your identity to be public for this peer review?** For information about this choice, including consent withdrawal, please see our Privacy Policy

Reviewer #1: **Yes: ** Prof Dr Mithra N Hegde

---

## [Author Response · Author response to Decision Letter 2]

25 Nov 2024

Dear Reviewer,

We would like to express our sincere gratitude for your valuable comments, which have significantly helped improve the quality of our manuscript. In response, the mentioned reference was added to the methodology section of the manuscript.

Thank you for considering our revised manuscript.

Sincerely,

Zeinab Rafiee

---

## [Decision Letter · Decision Letter 2]

Dear Dr. Rafiee,

Thank you for submitting your manuscript to PLOS ONE. After careful consideration, we feel that it has merit but does not fully meet PLOS ONE’s publication criteria as it currently stands. Therefore, we invite you to submit a revised version of the manuscript that addresses the points raised during the review process.

Kindly read all the comments given by the reviewers carefully and address them; make the changes in the revised manuscript accordingly.

Best regards and keep well

We look forward to receiving your revised manuscript.

Kind regards,

Mohmed Isaqali Karobari, BDS, MScD.Endo, Ph.D. Endo, FDS, FPFA, MFDS

Academic Editor

PLOS ONE

Journal Requirements:

Additional Editor Comments:

Dear Authors,

Kindly read all the comments given by the reviewers carefully and address them; make the changes in the revised manuscript accordingly.

Best regards and keep well

Reviewers' comments:

Reviewer's Responses to Questions

**Comments to the Author**

Reviewer #3: All comments have been addressed

Reviewer #4: (No Response)

2. Is the manuscript technically sound, and do the data support the conclusions?

Reviewer #3: Yes

Reviewer #4: Partly

3. Has the statistical analysis been performed appropriately and rigorously?

Reviewer #3: Yes

Reviewer #4: Yes

4. Have the authors made all data underlying the findings in their manuscript fully available?

Reviewer #3: Yes

Reviewer #4: Yes

5. Is the manuscript presented in an intelligible fashion and written in standard English?

Reviewer #3: Yes

Reviewer #4: Yes

Reviewer #3: manuscript is well written and addressed all the procedures in detail. hence it can be accepted for publication

Reviewer #4: (No Response)

**Do you want your identity to be public for this peer review?** For information about this choice, including consent withdrawal, please see our Privacy Policy

Reviewer #3: No

Reviewer #4: **Yes: ** Prof. Dr. Faiza Awais

---

## [Author Response · Author response to Decision Letter 3]

22 Feb 2025

We would like to express our sincere gratitude for your valuable comments, which have significantly helped improve the quality of our manuscript. In response, we have thoroughly revised the manuscript and addressed all the concerns raised by the reviewers in a point-by-point manner. The changes have been highlighted in the revised manuscript.

Kind regards,

Zeinab Rafiee

---

## [Editor Report · Decision Letter 3]

Tooth discoloration caused by nanographene oxide as an irrigant and intracanal medicament in the endodontic treatment of extracted single-rooted teeth: an ex-vivo study

PONE-D-24-30140R3

Dear Dr. Rafiee,

We’re pleased to inform you that your manuscript has been judged scientifically suitable for publication and will be formally accepted for publication once it meets all outstanding technical requirements.

Kind regards,

Mohmed Isaqali Karobari, BDS, MScD.Endo, Ph.D. Endo, FDS, FPFA, FICD, MFDS

Academic Editor

PLOS ONE
---

## [Editor Report · Acceptance letter]

PONE-D-24-30140R3

PLOS ONE

Dear Dr. Rafiee,

I'm pleased to inform you that your manuscript has been deemed suitable for publication in PLOS ONE. Congratulations! Your manuscript is now being handed over to our production team.

Kind regards,

on behalf of

Prof Dr. Mohmed Isaqali Karobari

Academic Editor

PLOS ONE